

# Blood parasites in Passeriformes in central Germany: prevalence and lineage diversity of Haemosporida (*Haemoproteus*, *Plasmodium* and *Leucocytozoon*) in six common songbirds

Yvonne R. Schumm[1], Christine Wecker[1], Carina Marek[1], Mareike Wassmuth[1], Anna Bentele[1], Hermann Willems[2], Gerald Reiner[2] and Petra Quillfeldt[1]

[1] Department of Animal Ecology & Systematics, Justus Liebig University, Giessen, Germany
[2] Department of Clinical Veterinary Sciences, Justus Liebig University, Giessen, Germany

Corresponding author
Yvonne R. Schumm,
Yvonne.R.Schumm@bio.uni-giessen.de

## ABSTRACT

**Background**. Avian Haemosporida are vector-borne parasites that commonly infect Passeriformes. Molecular analyses revealed a high number of different lineages and lineage specific traits like prevalence and host-specificity, but knowledge of parasite prevalence and lineage diversity in wild birds in Central Germany is still lacking.

**Results**. Blood samples from a total of 238 adult and 122 nestling songbirds belonging to six species were investigated for infections with avian haemosporidian genera and lineages (*Haemoproteus* spp., *Plasmodium* spp., *Leucocytozoon* spp.) and *Trypanosoma avium* using PCR, targeting the parasite mitochondrial cytochrome b gene and 18S ribosomal RNA. In total, the prevalence in adult birds was 31.3% infected with *Haemoproteus*, 12.5% with *Plasmodium* and 71.0% with *Leucocytozoon* (nestlings excluded). None of the tested birds was infected with *Trypanosoma avium*. Only in two nestling birds, aged 12–17 days, a *Leucocytozoon* spp. infection was proven. Among 225 successfully sequenced samples, we found four *Haemoproteus*, three *Plasmodium* and 19 *Leucocytozoon* lineages, including two new *Leucocytozoon* lineages. Furthermore, we report two new host-lineage associations.

**Conclusions**. As first study investigating avian haemosporidian parasites in Central Germany, we provide new information on genetic diversity of Haemosporida infecting Passeriformes. We show that even with a small sample size new lineages as well as previously unknown linkages between certain lineages and host species can be detected. This may help to elucidate the diversity of lineages as well as lineage-host-connections of avian Haemosporida.

## INTRODUCTION

Avian blood parasites (Hematozoa) infect both domestic and wild birds, and therefore they have been objects of intensive scientific research over a long period (*Valkiūnas, 1996*; *Valkiūnas, 2005*; *Bensch et al., 2013*). Among the most common blood parasite

orders are the Haemosporida. These include among others the avian malaria-like genera *Haemoproteus* and *Leucocytozoon* as well as the pathogen of avian malaria *Plasmodium*. These avian Haemosporida belonging to the phylum Apicomplexa share a similar but complex life cycle (*Schmid et al., 2017*), including asexual stages of reproduction in a bird host and sexual stages within a vector (*Valkiūnas, 2005*; *Santiago-Alarcon, Palinauskas & Schaefer, 2012*). Another common blood parasite is *Trypanosoma avium*, flagellate protozoans living in the bloodstream of birds (*Hamilton, Gibson & Stevens, 2007*). Hematophagous dipterans are the vectors of these parasites. *Leucocytozoon* parasites are vectored by blood-sucking dipterans of two families: black flies (Simuliidae) and biting midges (Ceratopogonidae) (*Freund et al., 2016*; *Lotta et al., 2016*). Ceratopogonid midges are also vectors for *Haemoproteus* spp. (*Desser & Bennett, 1993*). The main vectors for *Plasmodium* spp. are several species of mosquitos (Culicidae) (*Medeiros, Hamer & Ricklefs, 2013*). The transmission for *T. avium* remains still unclear with various blood-sucking insects mentioned as possible vectors (*Votypka et al., 2002*). Haemosporida are generally considered as pathogens with a low pathogenicity and harmless in bird populations (*Wiersch et al., 2007*; *Ciloglu et al., 2016*), but several studies demonstrated different costs on life-history traits associated with Haemosporida infections: Haemosporidian parasites can affect the body condition (*Valkiūnas et al., 2006*), reproductive success (e.g., *Hunter, Rohner & Currie, 1997*; *Merino et al., 2000*; *Marzal et al., 2005*; *Tomás et al., 2007a*; *Knowles, Palinauskas & Sheldon, 2010*) and the survival (e.g., *Dawson & Bortolotti, 2000*; *Møller & Nielsen, 2007*; *Donovan et al., 2008*; *Bueno et al., 2010*).

Haemosporida are abundant in many avian families and occur worldwide except in Antarctica (*Valkiūnas, 2005*; *Bensch, Hellgren & Perez-Tris, 2009*; *Clark, Clegg & Lima, 2014*; *Vanstreels et al., 2014*). However, there are interspecific differences in the parasite prevalence (e.g., *Bennett, Bishop & Peirce, 1993*; *Bennett, Peirce & Ashford, 1993*; *Valkiūnas, 2005*) and in some orders, e.g., Passeriformes, the parasites are more abundant (*Valkiūnas, 2005*). On the basis of recent molecular studies, the diversity of Haemosporida species and lineages may be assessed more precisely. The species diversity seems to be as high as the diversity of avian species (*Bensch et al., 2004*) or even higher (*Schmid et al., 2017*) and thousands of lineages may exist (*Szollosi et al., 2011*). Lineages can differ from each other in only one single nucleotide (e.g., one substitution) of the mitochondrial cytochrome b gene (*Bensch et al., 2004*; *Hellgren, Waldenström & Bensch, 2004*; *Bensch, Hellgren & Perez-Tris, 2009*; *Chagas et al., 2017*). For the order Passeriformes alone, 912 *Haemoproteus*, 852 *Plasmodium* and 600 *Leucocytozoon* lineages are deposited in the avian haemosporidian parasite database MalAvi (*Bensch, Hellgren & Perez-Tris, 2009*; *MalAvi, 2018*). In addition, molecular surveys vastly improved our understanding about the host specificity of avian haemosporidian infections (*Ciloglu et al., 2016*). The Haemosporida occupy niches that vary from extreme host generalization to extreme host specialization (*Okanga et al., 2014*). Generally, *Plasmodium* spp. is known to be more host-generalized (e.g., *Waldenström et al., 2002*; *Križanauskiene et al., 2006*; *Dimitrov, Zehtindjiev & Bensch, 2010*; *Mata et al., 2015*), whereas species of *Leucocytozoon* as well as *Haemoproteus* are considered to be more host-specific (e.g., *Waldenström et al., 2002*; *Beadell et al., 2004*; *Forrester & Greiner, 2008*; *Dimitrov, Zehtindjiev & Bensch, 2010*; *Jenkins & Owens, 2011*). *Trypanosoma avium* seems

**Table 1   Numbers of sampled adult and nestling songbirds.** Sample sizes per sex and parasite genus are given (P, *Plasmodium* spp.; H, *Haemoproteus* spp.; L, *Leucocytozoon* spp.; T, *Trypanosoma avium*).

| Species | Year of sampling | Number of specimens (*n*) | | | | | |
|---|---|---|---|---|---|---|---|
| | | Adult (male/female) | | | | Nestling | |
| | | **P** | **H** | **L** | **T** | **L** | **T** |
| Blue tit (*C. caeruleus*) | 2017 | **58** (30/28) | **58** (30/28) | **58** (30/28) | **58** (30/28) | **65**[a] | **65**[a] |
| Great tit (*P. major*) | 2015/2018 | **32** (14/18) | **32** (14/18) | **142** (49/93) | **68** (18/50) | **57**[b] | **57**[b] |
| Coal tits (*P. ater*) | 2017 | **4** (2/2) | **4** (2/2) | **4** (2/2) | **4** (2/2) | 0 | 0 |
| Eurasian tree sparrow (*P. montanus*) | 2015/2017 | **10** (1/9) | **10** (1/9) | **10** (1/9) | **10** (1/9) | 0 | 0 |
| European pied flycatcher (*F. hypoleuca*) | 2015/2017 | **14** (2/12) | **14** (2/12) | **14** (2/12) | **14** (2/12) | 0 | 0 |
| Eurasian nuthatch (*S. europaea*) | 2015/2017 | **10** (3/7) | **10** (3/7) | **10** (3/7) | **10** (3/7) | 0 | 0 |

**Notes.**
[a] Age: 16 days.
[b] Age: 12–17 days.

to possess a relatively low host-specificity (*Bennett, Earle & Squires-Parsons, 1994*; *Sehgal, Jones & Smith, 2001*).

Furthermore, prevalence varies not merely interspecific but also with the age of the birds. A common pattern observed in host-parasite assemblages is a higher abundance of parasites in juvenile compared to adult birds (*Sol, Jovani & Torres, 2003*). However, for blue tits nesting in nest boxes, blood parasites are far less prevalent in nestlings than in adult birds (*Cosgrove et al., 2006*; *Martinez-de la Puente et al., 2013*). Possibly because box-nesting species may be shield from vector exposure due to their enclosed surroundings (*Dunn et al., 2017*).

The aims of the present study were (i) to assess the prevalence of the Haemosporida: *Plasmodium*, *Haemoproteus* and *Leucocytozoon* as well as *T. avium* in six species of wild Passeriformes in Central Germany, (ii) to identify and compare the lineage diversity among the birds and to record interspecific-shared lineages by means of mitochondrial cytochrome b sequencing as well as (iii) to compare the prevalence in adult and nestling birds.

## MATERIAL & METHODS

### Origin and preparation of the samples

Bird capture and sampling were carried out under license (Animal welfare officer of the University of Giessen, no. 662_GP and 828_GP, and the Regierungspräsidium Giessen, no. 109-2012 and 77-2016) in accordance with the German legislation. Blood samples from 360 Passeriformes of four families and six species (blue tit *Cyanistes caeruleus*, Paridae; great tit *Parus major*, Paridae; coal tit *Periparus ater*, Paridae; eurasian tree sparrow *Passer montanus*, Passeridae; european pied flycatcher *Ficedula hypoleuca*, Muscicapidae and eurasian nuthatch *Sitta europaea*, Sittidae) were collected during April to June in the years 2015, 2017 and 2018 (Table 1).

Sample sites were located in and closely around the city Giessen (50°35′2.584″N 8°40′42.251″E, Hesse, Central Germany). All birds were captured at their nest boxes by hand and blood-sampled by brachial venipuncture. The blood was stored on Whatman

**Table 2  Primer pairs and their PCR conditions used for blood parasite screening.**

| Primer pair | Fragment size (bp) | Initial denaturation | Denaturation annealing extension | Cycles | Final extension | Target gene |
|---|---|---|---|---|---|---|
| HaemF | | | 30 s/94 °C | | | |
| | 480 | 3 min/94 °C | 30 s/55 °C | 35 | 10 min/72 °C | Cytochrome b |
| HaemR2 | | | 45 s/72 °C | | | |
| HaemFL | | | 30 s/94 °C | | | |
| | 600 | 3 min/94 °C | 30 s/51 °C | 35 | 10 min/72 °C | Cytochrome b |
| HaemNR3 | | | 45 s/72 °C | | | |
| TryF | | | 15 s/95 °C | | | |
| | 122 | 10 min/95 °C | 30 s/56 °C | 40 | 10 min/72 °C | 18S rRNA |
| TryR | | | 60 s/72 °C | | | |

FTA classic cards (Whatman®, UK). For DNA isolation a 3 × 3 mm piece of each sample was cut out of the FTA card with a sterile scalpel blade. Subsequently the DNA was extracted according to the ammonium-acetate protocol by *Martinez et al. (2009)* and purified with NZYspintech-columns (NZYTech, Lda.-Genes & Enzymes, Portugal) or Zymo-Spin™ IIC columns (Zymo Research, USA). The presence and concentration of DNA were confirmed and determined with NanoDrop2000c UV-Vis Spectrophotometer (NanoDrop Technologies, USA). If the DNA concentration was higher than 80 ng/µl, samples were diluted to 30 ng/µl.

## Parasite screening

A partial amplification of the mitochondrial cytochrome b gene of the different Haemosporida was accomplished by polymerase chain reaction (PCR) with the respective primers (Table 2).

For the detection of *Haemoproteus/Plasmodium* species the primer pair HaemF (5′-ATGGTGCTTTCGATATATGCATG 3′) and HaemR2 (5′-GCATTATCTGGATGTGATAA TGGT-3′) (*Bensch et al., 2000*) was used. For *Leucocytozoon* detection we applied primer pair HaemFL (5′- ATGGTGTTTTAGATACTTACATT-3′) and HaemNR3 (5′-ATAGAAAGATAAGAAATACCATTC-3′) (*Hellgren, Waldenström & Bensch, 2004*). To test for *T. avium* infections the accomplished primer pair was TryF (5′-ATGCACTAGGCACCGTCG-3′) and TryR (5′-GGAGAGGGAGCCTGAGAAATA-3′) (*Martinez-de la Puente et al., 2013*) targeting the 18S ribosomal RNA (Table 2).

Nestlings were only checked for possible *Leucocytozoon* spp. and *T. avium* infections. The PCR reactions with HaemF and HaemR2 consisted of 20 µl reaction volumes containing 2 µl DNA template (14.2–80 ng/µl), 1.2 µl of each primer (10 µM), 10 µl of InnuMix PCR Master Mix (2×, Analytik Jena AG, Germany) and 5.6 µl nuclease-free water. The PCR reaction for the detection of *Leucocytozoon* spp. contained 2.5 µl DNA template (8.0–80 ng/µl), 0.6 µl of each primer (20 µM), 10.6 µl InnuMix PCR Master Mix and 5.7 µl nuclease-free water. For *T. avium* evidence the PCR reaction volume was 20 µl consisting of 2.5 µl DNA template, 0.6 µl of each primer (20 µM), 10.6 µl DreamTaq PCR Master Mix (Thermo Fisher Scientific, Germany) and 5.7 µl nuclease-free water. The PCR reaction

parameters using thermal cyclers peqSTAR 96Q (Peqlab, Germany) and Tone (Biometra, Germany) are given in Table 2. A positive as well as a negative control were included in each run to ensure PCR was working properly. PCR amplicons were visualized using QIAxcel Advanced (Qiagen, Switzerland) high-resolution capillary gel electrophoresis. Samples rendering a clear peak during gel electrophoresis were bi-directional Sanger sequenced by Microsynth-Seqlab (Sequence Laboratories Goettingen GmbH, Germany).

## Phylogenetic and statistical analyses

The forward and reverse sequences were assembled and trimmed in CLC Main Workbench 7.6.4 (CLC Bio, Qiagen, Denmark). PCR and sequencing were repeated, if not all nucleotides of a sequence could be determined unambiguously. Sequences were excluded from network construction when repetitions did not improve sequence quality.

The consensus sequences were assigned to a parasite lineage by BLAST (BLASTN 2.3.0 +, *Zhang et al., 2000*) using the database MalAvi (*Bensch, Hellgren & Perez-Tris, 2009*). Constructions of lineage networks for each Haemosporida genus, using the median-joining network method, were performed with PopART 1.7 (*Bandelt, Forster & Röhl, 1999*; *Leigh & Bryant, 2015*) after aligning the sequences in BioEdit v7.2.5 (*Hall, 1999*). For these alignments, we used 123 *Leucocytozoon* (478 bp), 39 *Haemoproteus* (463 bp) and 14 *Plasmodium* consensus sequences (440 bp).

Statistical evaluation of comparing prevalences and number of interspecific shared lineages per genus was performed with R (*R Core Team, 2016*) using the R Package R commander. To compare the equality of proportions of parameters mentioned above the frequency distribution test Pearson's Chi$^2$-test was applied. A significance level of $p < 0.05$ was used.

# RESULTS

## Blood parasite prevalence

We detected three Haemosporida parasite genera in the local bird population in and around Giessen. The overall prevalence was 31.3% for *Haemoproteus*, 12.5% for *Plasmodium* and 71.0% for *Leucocytozoon* (nestlings excluded) (Table 3). Only blue tits and great tits were infected with all three genera. Coal tits were infected with *Plasmodium* and *Leucocytozoon* and tree sparrows with *Plasmodium* and *Haemoproteus*. Nuthatches showed infections with *Haemoproteus* and *Leucocytozoon*. For pied flycatchers we only found evidence of *Haemoproteus* infections (Table 3).

Prevalence rates of *Leucocytozoon* spp. were highest in blue tits (94.8%). The highest prevalence of *Plasmodium* infections occurred in coal tits (25.0%). *Haemoproteus* was almost equally common in blue tits (36.2%) and great tits (40.6%) (Table 3). The prevalence for the three haemosporidian genera in adult birds was different between the genera (Pearson's Chi$^2$-test: $\chi^2 = 129.1$, $df = 2$, $p < 0.001$). Comparing the prevalence of *Haemoproteus* and *Plasmodium*, *Haemoproteus* spp. was significantly more prevalent (Pearson's Chi$^2$-test: $\chi^2 = 13.2$, $df = 1$, $p < 0.001$), and *Leucocytozoon* was significantly more prevalent than the other two genera (Pearson's Chi$^2$-test: *Haemoproteus*/*Leucocytozoon*: $\chi^2 = 53.7$, $df = 1$, $p < 0.001$; *Plasmodium*/*Leucocytozoon*:
**Table 3 Haemosporida and *Trypanosoma avium* prevalence in six songbird species.**

| Species (No. of samples) | *Haemoproteus* spp. | | *Plasmodium* spp. | | *Leucocytozoon* spp. | | *Trypanosoma avium* | |
|---|---|---|---|---|---|---|---|---|
| | positive/ negative | Prevalence (%) | positive/ negative | Prevalence (%) | positive/ negative | Prevalence (%) | positive/ negative | Prevalence (%) |
| *C. caeruleus* | | | | | | | | |
| Adult (58) | 21/37 | **36.2** | 6/52 | **10.3** | 55/3 | **94.8** | 0/58 | **0.0** |
| Nestling (65) | | | | | 0/65 | **0.0** | 0/65 | **0.0** |
| *P. major* | | | | | | | | |
| Adult (142)[a] | 13/19 | **40.6** | 7/25 | **21.9** | 110/32 | **77.5** | 0/68 | **0.0** |
| Nestling (57) | | | | | 2/55 | **3.5** | 0/57 | **0.0** |
| *P. ater* | | | | | | | | |
| Adult (4) | 0/4 | **0.0** | 1/3 | **25.0** | 1/3 | **25.0** | 0/4 | **0.0** |
| *P. montanus* | | | | | | | | |
| Adult (10) | 1/9 | **10.0** | 2/8 | **20.0** | 0/10 | **0.0** | 0/10 | **0.0** |
| *F. hypoleuca* | | | | | | | | |
| Adult (14) | 3/11 | **21.4** | 0/14 | **0.0** | 0/14 | **0.0** | 0/14 | **0.0** |
| *S. europaea* | | | | | | | | |
| Adult (10) | 2/8 | **20.0** | 0/10 | **0.0** | 3/7 | **30.0** | 0/10 | **0.0** |
| **Total** | 40/88 | **31.3** | 16/112 | **12.5** | 171/189 | **47.5** | 0/286 | **0.0** |
| **Nestlings excluded** | | | | | 169/69 | **71.0** | 0/164 | **0.0** |

Notes.

[a] For *Haemoproteus* spp. and *Plasmodium* spp. the sample size was 32 and for *Trypanosoma avium* 68 adult great tits.

$\chi^2 = 114.0$, $df = 1$, $p < 0.001$). For none of the tested birds a *T. avium* infection was detected. In blue tit nestlings a haemosporidian infection could not be detected by PCR. In great tit nestlings we proved an infection with *Leucocytozoon*, although the prevalence with 3.5% tits infected (2 out of 57 nestlings) was low.

## Lineage diversity

In total, 26 haemosporidian lineages could be identified in the MalAvi database. We found four *Haemoproteus* spp., three *Plasmodium* spp. and 19 *Leucocytozoon* spp. lineages (Table 4). Great and blue tits showed similar high lineage diversities ($n = 15$ lineages: 2 *Haemoproteus*, 1 *Plasmodium* and 12 *Leucocytozoon* lineages; $n = 17$: 1 *Haemoproteus*, 3 *Plasmodium* and 13 *Leucocytozoon* lineages, respectively). We identified two new *Leucocytozoon* lineages (CYACAE02, GenBank accession number MH758695, and CYACAE03, MH758696), infecting blue tits which differ each in 0.21% (1 nucleotide) from their closest MalAvi match. For new lineages PCR and sequencing were performed twice to verify the results.

The lineage diversity of the other four species was less pronounced probably due to smaller sample sizes. For pied flycatchers we detected one *Haemoproteus* lineage (PFC1). For tree sparrows we found three lineages (1 *Haemoproteus*: PADOM03 and 2 *Plasmodium*: GRW11 and SGS1), for coal tits one lineage each for *Plasmodium* (TURDUS1) and *Leucocytozoon* (PARUS19) and for nuthatches one *Haemoproteus* (PARUS1) and two *Leucocytozoon* lineages (PARUS7 and PARUS20).

**Table 4** Parasite lineages found in the six songbird species with their closest MalAvi match, the respective accession number and query cover in %.

| Parasite | Lineage (MalAvi) | Accession number (GenBank) | *n* | Match (%) | Host species (Number of individuals) | Lineage prevalence (%)[a] | Reference accession number |
|---|---|---|---|---|---|---|---|
| *Haemoproteus*[b] | PARUS1 | JQ778282 | 34 | 100 | *C. caeruleus* (21) *P. major* (11) *S. europaea* (2) | 15.1 | *Glaizot et al. (2012)* |
| *Haemoproteus* | PADOM03 | KJ488647 | 1 | 100 | *P. montanus* (1) | 0.4 | *Drovetski et al. (2014)* |
| *Haemoproteus* | PHSIB1 | KJ396634 | 1 | 100 | *P. major* (1) | 0.4 | *Scordato & Kardish (2014)* |
| *H. pallidus* | PFC1 | JX026899 | 3 | 100 | *F. hypoleuca* (3) | 1.3 | *Valkiūnas et al. (2013)* |
| *Plasmodium* | GRW11 | AY831748 | 3 | 100 | *C. caeruleus* (2) *P. montanus* (1) | 1.3 | *Perez-Tris & Bensch (2005)* |
| *Plasmodium* | SGS1 | AB542064 | 11 | 99–100 | *C. caeruleus* (3) *P. major* (7) *P. montanus* (1) | 4.9 | *Ejiri et al. (2011)* |
| *P. circumflexum* | TURDUS1 | KP000842 | 2 | 100 | *C. caeruleus* (1) *P. ater* (1) | 0.9 | *Ciloglu et al. (2016)* |
| *Leucocytozoon*[c] | PARUS4 | KJ488615 | 11 | 99–100 | *P. major* (10) *C. caeruleus* (1) | 4.9 | *Mata et al. (2015)* |
| *Leucocytozoon* | PARUS7 | KJ488817 | 3 | 97–100 | *P. major* (1) *S. europaea* (2) | 1.3 | *Mata et al. (2015)* |
| *Leucocytozoon* | PARUS11 | HM234019 | 5 | 99–100 | *C. caeruleus* (5) | 2.2 | *Jenkins & Owens (2011)* |
| *Leucocytozoon* | PARUS12 | HM234020 | 3 | 100 | *C. caeruleus* (3) | 1.3 | *Jenkins & Owens (2011)* |
| *Leucocytozoon* | PARUS13 | HM234021 | 2 | 100 | *C. caeruleus* (2) | 0.8 | *Jenkins & Owens (2011)* |
| *Leucocytozoon* | PARUS14 | HM234022 | 15 | 100 | *C. caeruleus* (15) | 6.7 | *Jenkins & Owens (2011)* |
| *Leucocytozoon* | PARUS15 | HM234023 | 1 | 100 | *C. caeruleus* (1) | 0.4 | *Jenkins & Owens (2011)* |
| *Leucocytozoon* | PARUS16 | HM234024 | 14 | 100 | *P. major* (14) | 6.2 | *Jenkins & Owens (2011)* |
| *Leucocytozoon* | PARUS18 | HM234026 | 6 | 99–100 | *C. caeruleus* (2) *P. major* (4) | 2.7 | *Jenkins & Owens (2011)* |
| *Leucocytozoon* | PARUS19 | HM234027 | 67 | 99–100 | *C. caeruleus* (13) *P. major* (53) *P. ater* (1) | 29.8 | *Jenkins & Owens (2011)* |
| *Leucocytozoon* | PARUS20 | KJ488629 | 4 | 99–100 | *P. major* (3) *S. europaea* (1) | 1.8 | *Mata et al. (2015)* |
| *Leucocytozoon* | PARUS22 | HM234031 | 2 | 100 | *P. major* (1) *C. caeruleus* (1) | 0.9 | *Jenkins & Owens (2011)* |
| *Leucocytozoon* | PARUS33 | JX867108 | 2 | 100 | *P. major* (2) | 0.9 | *Van Rooyen et al. (2013b)* |
| *Leucocytozoon* | PARUS34 | JX855049 | 2 | 100 | *P. major* (2) | 0.9 | *Van Rooyen et al. (2013b)* |
| *Leucocytozoon* | PARUS72 | KJ488759 | 3 | 99 | *P. major* (3) | 1.3 | *Mata et al. (2015)* |
| *Leucocytozoon* | PARUS74 | KJ488766 | 9 | 100 | *P. major* (6) *C. caeruleus* (3) | 4.0 | *Mata et al. (2015)* |
| *Leucocytozoon* | PARUS84 | KJ488911 | 6 | 100 | *C. caeruleus* (6) | 2.7 | *Mata et al. (2015)* |

*(continued on next page)*
Table 4 (*continued*)

| Parasite | Lineage (MalAvi) | Accession number (GenBank) | n | Match (%) | Host species (Number of individuals) | Lineage prevalence (%)[a] | Reference accession number |
|---|---|---|---|---|---|---|---|
| *Leucocytozoon* | CYACAE02 | MH758695 | 1 | 100 | *C. caeruleus* (1) | 0.4 | New lineage |
| *Leucocytozoon* | CYACAE03 | MH758696 | 3 | 98–100 | *P. major* (1) *C. caeruleus* (2) | 1.3 | New lineage |

**Notes.**
[a] Percentage of each lineage among all infected birds (*n* = 225).
[b] One sample could be determined as *Haemoproteus* spp., but could not be assigned to one certain lineage by BLAST against the MalAvi database due to an insufficient sequence quality.
[c] 10 samples could be determined as *Leucocytozoon* spp., but could not be assigned to one certain lineage by BLAST against the MalAvi database due to an insufficient sequence length and quality.

Several sequences (*n* = 11) could not be assigned to a single reference sequence due to an insufficient sequence length or quality. In these cases, the sequences were determined to the closest related reference lineages (Table S1).

## Host specificity

Maximally three out of six Passeriformes species were found infected with the same lineage (Table 4). We recorded the highest number of lineages occurring in several host species for *Plasmodium*. According to the BLAST results, all identified *Plasmodium* lineages occurred in more than one species (Table 4). In contrast, smaller percentages of lineages infecting more than one host species, were found for *Leucocytozoon* (36.8% of the lineages infected two host species and only 5.3% (one lineage: PARUS19) infected three host species) and *Haemoproteus* with 25% (only one lineage (PARUS1) infected more than one host species) (Table 4). The difference in the percentage of interspecific shared lineages was not significant between the three Haemosporida genera (Pearson's Chi$^2$-test: $\chi^2 = 6.82$, $df = 4$, $p = 0.15$).

## Lineage networks

The networks for each haemosporidian genus revealed the occurrence of four lineages for *Haemoproteus* (Fig. 1), three for *Plasmodium* (Fig. 2) and 18 for *Leucocytozoon* (Fig. 3). In the *Leucocytozoon* network sequences from the samples NK17_P01 (*C. caeruleus*) and NK17_Y40 (*C. caeruleus*) occurred each separately from all the other sequences and had a BLAST of maximum 99% match with lineages deposit in MalAvi, indicating a new lineage (named CYACAE02 and CYACAE03). The genetic divergence to the closest MalAvi match of CYACAE02 (sample NK17_P01) is 0.21% (1 nucleotide difference to PARUS22, HM234031). The difference of CYACAE03 (sample NK17_Y40) to its closest related match PARUS19 (99% match in 477 bp) is 0.21%. The genetic divergence for the *Leucocytozoon* lineages in the network is 0.21–7.95%. All found lineages of the dataset are separated at least by one mutation (equivalent to one hatch mark in the network). With the exception of PARUS19 and PARUS74 (Fig. 3). The mutation differentiating these lineages is located in the cytochrome b gene before the fragment we used for network construction.

The *Haemoproteus* network (Fig. 1) shows that the lineage PFC1 (*H. pallidus*), occurring in all three positively tested pied flycatchers, and the lineage PADOM03, found in one tree sparrow, are clearly separated from the other *Haemoproteus* spp. lineages. *Haemoproteus*
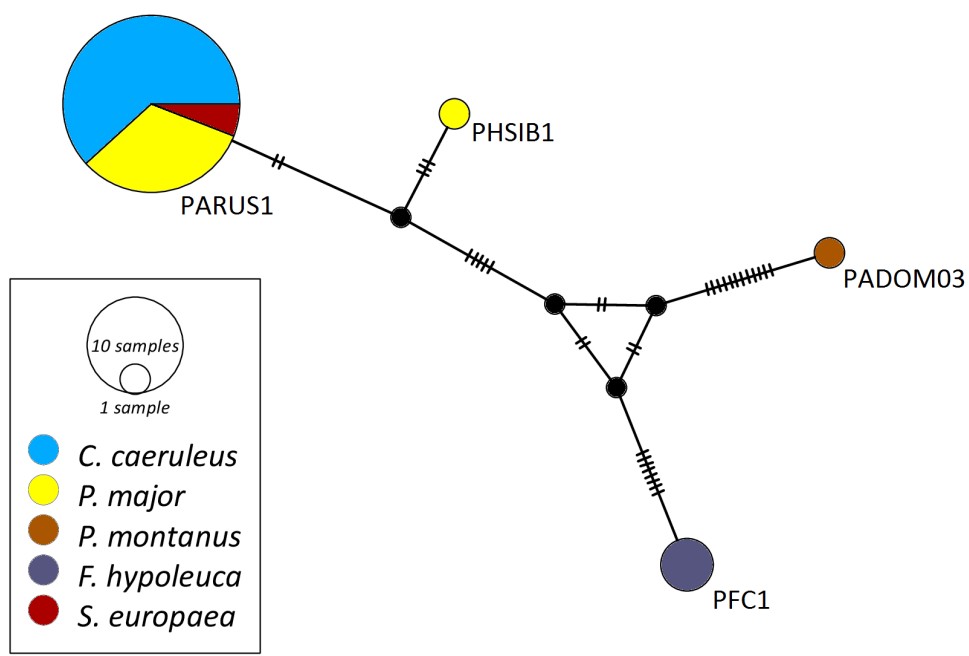

**Figure 1** **Median-joining network of mitochondrial cytochrome b gene lineages of *Haemoproteus* spp. (*n* = 39, 463 bp fragment).** Circles represent lineages, and the circle sizes are proportional to the lineage frequencies in the population. One hatch mark represents one mutation. Sampled host species are represented by different colors. Lineage names are noted at the associated circles.

lineages have a genetic divergence of 1.08–5.18%. The *Plasmodium* network (Fig. 2) illustrates the three lineages (SGS1, GRW11 and TURDUS1) found in this study. The results of this network lead to the assumption that TURDUS1 exclusively infects coal tits. However, it must be noted that also a blue tit was infected with TURDUS1, but the sample sequence was too short to be used for the alignment. *Plasmodium* lineages in the network range from 0.23–4.32% in their genetic divergence.

## DISCUSSION

### Prevalence in nestling birds

Contrary to a high percentage of adult blue and great tits infected with at least one haemosporidian parasite we did not detect a single haemosporidian or *T. avium* infection in nestling blue tits and a very low infection rate of *Leucocytozoon* spp. in great tit nestlings. Generally, nestlings should be highly susceptible to vector-borne infection diseases due to their confinement to the nest, nakedness and immunological naivety during nestling period (*Baker, 1975*). Possibly, more nestlings might have been infected in our study, but our methodology did not accomplish detection if the disease was still at the prepatent stage. Infections cannot be detected immediately after transmission because of the prepatency period (i.e., the period between initial infection and the release of gametocytes into the peripheral blood). We can distinguish between parasites with shorter prepatency periods like *Leucocytozoon* (5 to 6 days) (*Desser & Bennett, 1993*) or *T. avium* (24 to 48 h) (*Bennett,*

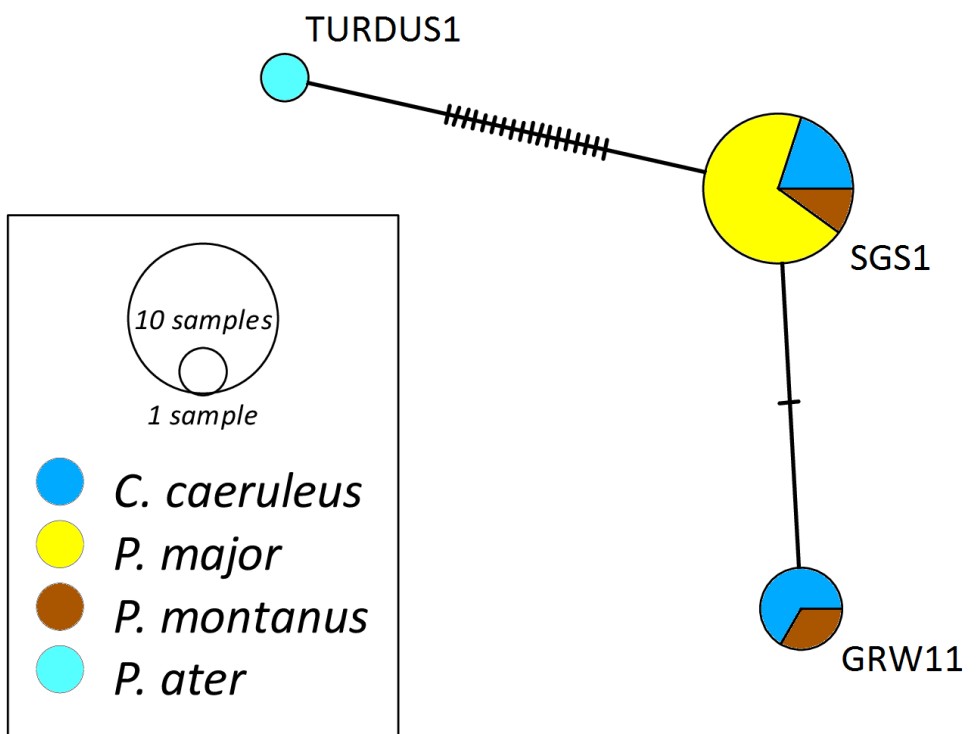

**Figure 2** Median-joining network of mitochondrial cytochrome b gene lineages of *Plasmodium* spp. (*n* = 14, 440 bp fragment). Circles represent lineages, and the circle sizes are proportional to the lineage frequencies in the population. One hatch mark represents one mutation. Sampled host species are represented by different colors. Lineage names are noted at the associated circles.

*1970*), and with longer prepatency periods like *Haemoproteus* or *Plasmodium*. The period for *Haemoproteus* spp. varies from 11 to 21 days, for *Plasmodium* it can last from few days up to more than one month until an infection is detectable in the peripheral blood, depending on host and parasite species (*Valkiūnas, 2005*; *Cosgrove et al., 2006*). Alternatively, it could simply mean that the nestlings had not been infected yet.

To test if an infection was transmitted already, it would have been necessary to remove nestlings from their nests, raising them in vector-free cages and checking regularly for subsequent development of patent infections. *Valkiūnas (2005)* applied this approach in chaffinch (*Fringilla coelebs*) nestlings. In his study, only two out of 67 chicks (3%), removed from the nest at 6 to 12 days of age, subsequently developed infections. Contrary, the infection rate of 25 to 50 day old wild fledglings was 36.2%, suggesting that most infections occurred after the nestlings had left the nest (*Valkiūnas, 2005*). The very low rate of infections in nestling tits in our study might also result from a lack of vector activity during the nestling period as in northern temperate climes dipteran vector populations reach their peak not until late summer (*Beaudoin et al., 1971*).

## Prevalence in adult birds

Within Germany, only a few studies deal with the prevalence and distribution of infections with avian haemosporidian parasites (microscopic examination: *Haberkorn, 1984*; *Krone*

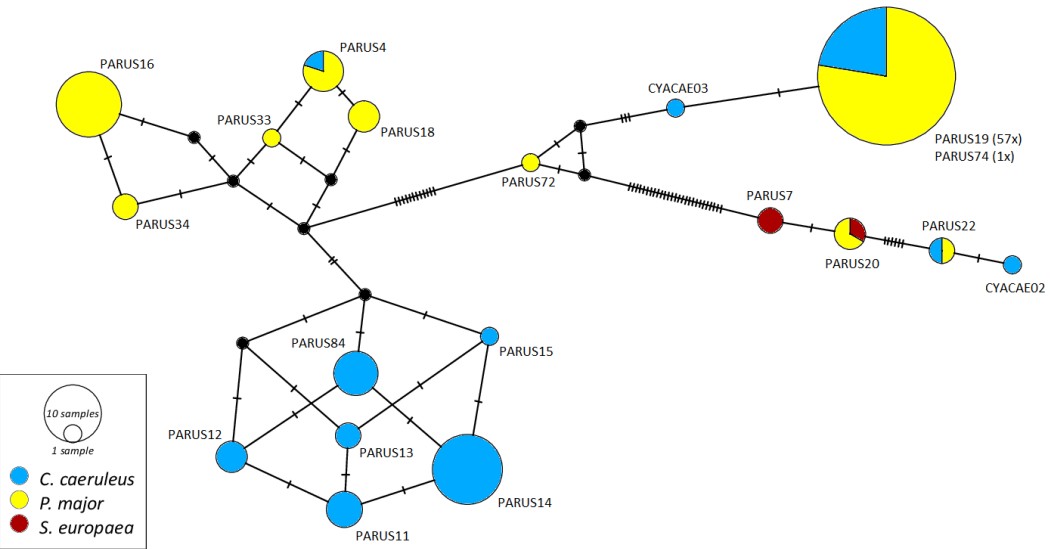

**Figure 3** Median-joining network of mitochondrial cytochrome b gene lineages of *Leucocytozoon* spp. (*n* = 123, 478 bp fragment). Circles represent lineages, and the circle sizes are proportional to the lineage frequencies in the population. One hatch mark represents one mutation. Sampled host species are represented by different colors. Lineage names are noted at the associated circles.

*et al., 2001*; PCR-based methods: *Wiersch et al., 2007*; *Jenkins & Owens, 2011*; *Santiago-Alarcon et al., 2016*) (see Table 5 for an overview of prevalences in great and blue tits sampled in Germany). *Wiersch et al. (2007)* sampled birds in the northern part of Germany. Infection prevalences in great tits were 30.4% for *Haemoproteus* and 46.4% for *Plasmodium*. In line with the present study, coal tits had no infections with *Haemoproteus* spp. and a similar infection rate with *Plasmodium* spp. (18.7%). The infection rate for pied flycatchers with *Haemoproteus* spp. (0.9%) was less than in the present study (21.4%). In contrast to our findings, *Wiersch et al. (2007)* reported *Plasmodium* infections in pied flycatchers (5.9%). *Santiago-Alarcon et al. (2016)* found no haemosporidian infection in eurasian nuthatches in Germany (20.0% *Haemoproteus* spp. and 30.0% *Leucocytozoon* spp. in the present study) and *Haberkorn (1984)* found no haemosporidian infections in coal tits (25.0% *Plasmodium* spp. and 25.0% *Leucocytozoon* spp. in this study). As far as we know, no other data are available for european tree sparrow infection rates in Germany.

In contrast to our results, low infection rates in blood smears from great tits were reported by *Haberkorn (1984)* (*Haemoproteus* spp.: 7.3%, *Leucocytozoon* spp.: 1.2%). These variations in prevalences might be based on methodological differences.

*Jenkins & Owens (2011)* recorded 33% *Leucocytozoon* prevalence in South Germany for great tits and 50% for blue tits, which is much lower than the prevalences reported here for the two species (77.3% and 94.8%, respectively). However, some studies report even higher prevalences especially for the avian malaria pathogen *Plasmodium* (e.g., 91% for blue tits from Switzerland, *Glaizot et al., 2012*; 100% in a blue tits population in England, *Szollosi et al., 2011*) and for *T. avium* (e.g., 49% of infected blue tits in Spain, *Fargallo & Merino, 2004*; 40% in female blue tits in Spain, *Tomás et al., 2007b*).

**Table 5  Overview of publications dealing with Haemosporida and *Trypanosoma avium* prevalences in great and blue tits sampled in Germany.** Research method, sample sizes, study region and prevalences of the different blood parasites are given (P, *Plasmodium* spp.; H, *Haemoproteus* spp.; L, *Leucocytozoon* spp.; T, *Trypanosoma avium*). NT, Not tested in the listed study.

| Reference | Study region | Method | Species (sample size) | Prevalence P | Prevalence H | Prevalence L | Prevalence T |
|---|---|---|---|---|---|---|---|
| *Haberkorn (1984)* | Western and northern Germany | Microscopic examination | *P. major* (82) | 2.4% | 8.5% | 1.2% | 0.0% |
| | | | *C. caeruleus* (66) | 0.0% | 7.6% | 1.5% | 0.0% |
| *Wiersch et al. (2007)* | Northern Germany | PCR-based | *P. major* (56) | 46.4% | 30.4% | NT | NT |
| *Jenkins & Owens (2011)* | Southern Germany | PCR-based | *P. major* (33) | NT | NT | 33.3% | NT |
| | | | *C. caeruleus* (24) | NT | NT | 41.6% | NT |
| *Santiago-Alarcon et al. (2016)* | Southwestern Germany | PCR-based | *P. major* (43) | | 46.5%[a] | | NT |
| | | | *C. caeruleus* (6) | | 0.0%[a] | | NT |
| Present study | Central Germany | PCR-based | *P. major* (142)[b] | 21.9% | 40.6% | 94.8% | 0.0% |
| | | | *C. caeruleus* (58) | 10.3% | 36.2% | 77.5% | 0.0% |

**Notes.**

[a] Only overall prevalence for the three Haemosprida was given in the publication (20 from 43 great tits were infected); no prevalences for the single genera were mentioned.

[b] For *Haemoproteus* spp. and *Plasmodium* spp. the sample size was 32 and for *Trypanosoma avium* 68 adult great tits.

However, it should be considered that sensitive PCR-based diagnostics are able to detect sporozoites of *Leucocytozoon* in the peripheral blood (*Valkiūnas et al., 2009*). Sporozoites are transmitted to the bird during the blood meal of the vector fly. As it is unclear whether all of these sporozoites result in an actual infection of the host, the evidence of haemosporidian lineages by PCR based method does not necessarily allow the conclusion that the parasites complete their entire life cycle in the host (*Valkiūnas et al., 2009*).

The high prevalence of *Leucocytozoon* spp. in blue and great tits in this study compared to the parasite genera *Plasmodium* and *Haemoproteus* may be associated with the vector abundance and behavior. Dipteran vectors of some Haemosporida genera may be more strictly ornithophilic than Culicidae vectors of *Plasmodium* spp., which feed on a broader range of vertebrates, reducing their potential for transmitting diseases to birds (*Savage et al., 2009*). The reasons for the high *Leucocytozoon* prevalence in our study area Hesse compared to other parts of Germany are speculative. One reason might be that Hesse is the most richly forested of all German states (42% of the state area are forests) (*Forest report Hesse, 2015*). Moreover, these forest sites are mostly near-natural and with a lot of woodland-running-waters, that are mostly (70%) in a good ecological condition (*Forest report Hesse, 2015*). As Simuliidae, the vectors of all *Leucocytozoon* species (except *L. caulleryi*, that is vectored by a biting midge; *Lotta et al., 2016*), need running waters for reproduction (*Lacey & Merritt, 2003*) the conditions for their reproduction in Hesse are good according to the habitat parameters mentioned above. However, we did not record any habitat parameters during our study and similarly no nationwide data throughout Germany on black fly distribution is available at the moment. Therefore, it is not possible to test regional prevalence of *Leucocytozoon* depending on the distribution and density of black flies so far.

Unfortunately, many studies do not consider all three avian haemosporidian genera and especially *Leucocytozoon* is underrepresented in the literature (*Van Rooyen et al., 2013a*).

Hence, a detailed comparison of prevalences in different avian hosts is difficult to assess. However, comparison of our results with studies from Germany and Europe show that the prevalences of avian haemosporidian infections differ among local bird populations (e.g., *Haberkorn, 1984*; *Wiersch et al., 2007*; *Santiago-Alarcon et al., 2016*). The factors causing these local differences are still poorly understood. *Szollosi et al. (2011)* showed that the distribution and prevalence of avian malarial parasite species are influenced by multiple factors, such as host and dipteran vector density, habitat characteristics or climatic conditions (see also *Wood et al., 2007*; *Merino et al., 2008*). Moreover, prevalence seems to be a lineage-specific trait (*Szollosi et al., 2011*). That shows the importance to investigate not only prevalence for the parasite genera, but rather identify the different parasite lineages infecting regional bird populations.

## Lineage diversity and host specificity

By using molecular phylogeny, we detected different lineages for each parasite genus. Few lineages differed by only one nucleotide, resulting in low genetic divergences. Other authors (e.g., *Bensch et al., 2000*; *Chagas et al., 2017*) also found low sequence divergences in Haemosporida. In this study, the two new *Leucocytozoon* lineages (CYACAE02 and CYACAE03) differ in one nucleotide each from their closest matching lineage, indicating they may have diverged only recently (*Bensch et al., 2000*).

The lineage PFC1 (*H. pallidus*), infecting only pied flycatchers in this study, is clearly separated in the network. It is possible that the infection with this lineage was transmitted outside Germany, as pied flycatchers, wintering in Africa, are the only long-distance migratory bird species in our study and infection with the PFC1 lineage might be vectored from dipteran vectors in Africa. However, the distribution of vectors on wintering and breeding grounds, especially for *Haemoproteus*, is poorly understood (*Dubiec et al., 2017*) and transmission of PFC1 lineage possibly also occurs in Europe (*Jones, 2017*).

Generally, host specificity of haemosporidian lineages differs among the genera with host specialists being predominant among *Haemoproteus* and *Leucocytozoon* lineages but being absent among *Plasmodium* lineages (*Mata et al., 2015*). This general pattern of host specificity, with *Haemoproteus* being the most host specialized and *Plasmodium* being more host generalized, is supported by several studies (e.g., *Ricklefs & Fallon, 2002*; *Waldenström et al., 2002*; *Križanauskiene et al., 2006*; *Dimitrov, Zehtindjiev & Bensch, 2010*; *Jenkins & Owens, 2011*; *Drovetski et al., 2014*; *Okanga et al., 2014*; *Mata et al., 2015*; *Ciloglu et al., 2016*). Less studies regarding host specificity of *Leucocytozoon* lineages exist, therefore it is necessary to do further investigations to confirm *Leucocytozoon* lineages to be, as assumed, host-specific mostly at avian order level and in some cases even on species level (*Forrester & Greiner, 2008*; *Ciloglu et al., 2016*). In the present study, we found no significant difference in the host specificity of the three haemosporidian genera. But due to small sample sizes and closely related host species (all within the order Passeriformes) general patterns should be proven with a higher number of samples and an increased range of species.

Our data complements existing knowledge about host specificity and distribution of some individual lineages as we obtained first records for lineages infecting a specific host species. The two *Leucocytozoon* lineages PARUS20 (isolated from sample NK17_C05,

MH758693) and PARUS7 (isolated from samples NK17_C04, MH758692 and NK17_C14, MH758694) were not detected in eurasian nuthatches (*S. europaea*) prior to this study (*Bensch, Hellgren & Perez-Tris, 2009*; *MalAvi, 2018*). This is also the first record of the PARUS7 lineage in the family Sittidae. Host specificity seems not to be determined by parasite genera but by the single lineages comparable with the lineage-specific prevalences. For example, several studies (e.g., *Ricklefs & Fallon, 2002*; *Beadell et al., 2009*; *Loiseau et al., 2012*) suggest that few lineages of avian malaria pathogen *Plasmodium* exhibit extreme generalization, whereas other lineages seem to be constrained to certain host families or even host species. Host shifts are often associated with a change in pathogen virulence (*Toft & Krater, 1990*). Therefore, invading a new host may increase or decrease parasite virulence (*Bull, 1994*). This might be the case for the two *Leucocytozoon* lineages (PARUS20 and PARUS7) infecting nuthatches.

## CONCLUSION

In summary, we found avian malaria and avian malaria-like pathogens of three genera (*Plasmodium, Haemoproteus* and *Leucocytozoon*) infecting common Passeriformes in Central Germany. The findings presented here provide knowledge about the distribution and prevalence of avian haemosporidian parasites in a geographic region, which has not yet been subject to studies investigating this kind of parasites. On the basis of a relatively small sample size we found numerous lineages and detected several first records of lineage infections as well as two new *Leucocytozoon* lineages. Comparison with studies from other parts of Germany pointed out regional differences in Haemosporida prevalence, in particular for *Leucocytozoon*. Understanding these patterns resulting in regional differences could be important in future to understand the epidemiology of blood parasites in wild bird populations.

## ACKNOWLEDGEMENTS

We would like to thank Wendy Gibson (School of Biological Sciences, University of Bristol) and Rasa Bernotienė (Nature Research Center, Vilnius) for trypanosome infected bird samples used as positive controls. We are grateful to academic editor Erika Braga and three anonymous reviewers for helpful comments on the manuscript. We thank all helpers of the nest box checks in 2015 (Jennifer Schwarz, David Ensslin and Carsten Hoth) and 2017/18 (Anna Bentele, Fabian Gausepohl, Benjamin Grünwald, Daniel Höhn and Michael Reis) and Tobias Warmann for labwork assistance.

### Funding

There was no external funding for this project.

### Competing Interests

The authors declare there are no competing interests.
## Author Contributions

- Yvonne R. Schumm conceived and designed the experiments, performed the experiments, analyzed the data, prepared figures and/or tables, authored or reviewed drafts of the paper, approved the final draft.
- Christine Wecker, Carina Marek and Anna Bentele performed the experiments, analyzed the data, approved the final draft.
- Mareike Wassmuth performed the experiments, approved the final draft.
- Hermann Willems performed the experiments, contributed reagents/materials/analysis tools, approved the final draft.
- Gerald Reiner conceived and designed the experiments, contributed reagents/materials/analysis tools, approved the final draft.
- Petra Quillfeldt conceived and designed the experiments, analyzed the data, contributed reagents/materials/analysis tools, authored or reviewed drafts of the paper, approved the final draft.

## Animal Ethics

The following information was supplied relating to ethical approvals (i.e., approving body and any reference numbers):

All sampling was performed in accordance to animal welfare standards, supervised by the Animal welfare officer of the University of Giessen and the Regierungspräsidium Giessen. The German equivalent to an IACUC number is the Intern number of the University: 662_GP and 828_GP.

## Field Study Permissions

The following information was supplied relating to field study approvals (i.e., approving body and any reference numbers):

Bird capture and sampling were carried out under a license from Regierungspräsidium Giessen: GI 15/8-Nr.109/2012 and GI 15/8-Nr.77/2016.

## Data Availability

Newly generated sequences are available in GenBank: MH758692, MH758693, MH758694, MH758695 and MH758696. The *Haemoproteus* and *Plasmodium* sequences are available in Table 4. The hosts and sites table is available at the MalAvi database: http://mbio-serv2.mbioekol.lu.se/Malavi/.

## Supplemental Information

Supplemental information for this article can be found online at http://dx.doi.org/10.7717/peerj.6259#supplemental-information.

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
