# Peer review of "Blood parasites in Passeriformes in central Germany: prevalence and lineage diversity of Haemosporida (Haemoproteus, Plasmodium and Leucocytozoon) in six common songbirds"

_PeerJ, doi:10.7717/peerj.6259_

## Round 0.1 · original submission · Major Revisions

The review process is now complete, and three thorough reviews from highly qualified referees are included at the bottom of this letter. All reviewers including myself agree the manuscript is well written and deserves to be published. Although there is considerable merit in your paper, we also identified some concerns that must be considered in your resubmission. I particularly agree with Reviewer #1 that the use of primers amplifying a short fragment of the parasite cyt b gene instead of those used to obtain comparable sequences deposited in MalAvi database need be justified.

Please, clarify the strategy to detect parasites using parasitological and molecular methods. To detect Trypanosoma avium in blood smears an examination at 400X is appropriate. Is this protocol used to confirm infection by Trypanosoma? How you cannot detect Leucocytozoon in blood smears considering a high molecular prevalence (around 70%) of such genus? Please, check the following information: "In general, Leucocytozoon and T. avium infections were not detected in blood smears." and discuss the issue accordingly.

Reviewer 1 ·

Basic reporting

This descriptive study provides additional information on the haemosporidian parasites prevalence and genetic diversity among six common passerine birds in central Germany. The birds were also investigated for Trypanosoma infections, but such were not found. The authors apply PCR-based and light microscopy methods to diagnose the infections. The manuscript have relatively good English but there are some parts of the text that have to be rephrased or clarified (L 4-6, L 111-112, L 140, L 180-182, L 201-203). There are also some incorrectly used terminology and scientific names. For instance, throughout the majority of the text, the authors used “lineage/s” for addressing the mitochondrial cytochrome b sequences, but in L 209-215 the “lineage” were replaced with “haplotype”. Please unify the terminology along the text or explain what is the difference between both terms in the context of your results. The correct spelling is order “Haemosporida” but not “Haemosporidia”! Please correct it along the text or use noncapital “haemosporidian/s parasites” when appropriate.
The Introduction section is well structured and provides good background related to the results of the study. However, I think that the purpose of collecting material from birds in central Germany is not clarified. What new the authors would expected to find in the avian blood parasites from this particular region? How your work fits to the broader field of knowledge? The statement on L 46-48 is a bit controversial as the prevalence of haemosporidian parasites in nestlings very much depends on the fledging period. When the fledging period is longer the nestlings are exposed longer to the potential vectors and the parasite have enough time to develop and be registered in the peripheral blood. Furthermore it depend if the birds species are hollow breeders and so on. In my opinion, the paragraph have to be extended with more information.

Experimental design

It is questionable why the authors did not use well established nested PCR protocol developed by Waldenstrom et al. 2004 and Hellgren et al. 2004. Nested PCR should increase the sensitivity of the method and in this case perhaps reflect in underestimated prevalence. Furthermore, one of the primer pairs (PaluF-PaluR) amplify shorter fragment of the mitochondrial cyt b gene, which result in an ambiguous identification of the haemosporidian cyt b lineages after BLAST in MalAvi database (Supplementary Table 1). Would you explain why this combination of primers was used?
It is not clear the number of samples screened with different primers. Obviously not all the samples were screened equally (L 86-88) for the potential parasite infection. In a descriptive study as this one, I think this could be a big flaw.

Validity of the findings

The results for the genera Haemoproteus and Plasmodium are expected and comparable with other studies in Europe. The prevalence for genus Leucocytozoon was the highest and it is surprising that none of the parasites were registered on the blood smears. Perhaps the usual light intensity of Leucocytozoon spp. in the wild birds prevent registration of these parasites by microscopic examination. An extensive microscopy would be required in order to confirm most of the infections. There are also indications that PCR-based methods could register sporozoites in the blood stream and abortive development of haemosporidian parasites (see Valkiunas et al. 2009 – doi: 10.1645/GE-2105.1, 2013 – doi: 10.1007/s00436-013-3375-6). I do not think that this is big flaw in the study, but I believe it worth consideration and discussion.
The conclusions (L 342-345) of the study I found quite irrelevant with the main findings. I would recommend that authors comment on how their data fit to the broader filed of knowledge then simply mention that “the impact of the pathogens on the avian fauna is difficult to estimate and future studies are necessary”.

Additional comments

I think that your manuscript might become a nice descriptive study if you use appropriate protocols and obtain good quality sequences. As it is now the identification of haemosporidian parasites is too ambiguous for simply describe the prevalence and the diversity in certain host community in Germany. Otherwise, some sort of hypothesis on the avian host specificity or ecological factors would be required. Please find attached a pdf file with some of the same and additional comments and suggestions on your manuscript.

Annotated reviews are not available for download in order to protect the identity of reviewers who chose to remain anonymous.

Reviewer 2 ·

Basic reporting

The manuscript is written in a correct English, and include sufficient background in the introduction using correct literature references.
However the manuscript is ambiguous to understood, it is difficult to follow what samples have been used for in each analyses because some samples were not used due to their length. This fact does that the work seems not to be designed with the objective indicate in the manuscript.
Although the Figures 2, 3, 4 and 5 are appropritately described an labeled, the Figure 1 is not clear. I think that is better if the authors design this one with color.
I think that the table 2 is not necessary.

Experimental design

Different factors indicate a not excellent experimental design of this manuscript.
I do not understand why the author did not check Plasmodium and Haemoproteus in Blue tit nestlings, or why great tit nestlings were not checked for Trypanosoma, or why blood smears of great tits were excluded.
The recollection of great tits samples in 2017 would be excellent to obtain more robust results.

The author could check if the results are similar using the sequences obtained using Primers HaemF and Haem R2 for Plasmodium/Haemoproteus and Primers HaemF2 and HaemR2L for Leucocytozoon and removing the samples of 2015.

Validity of the findings

I think that the authors could do a more novelty manuscript if the authors discuss deeper why is important to study blood parasites in central Germany in comparison with other works about parasites in Germany. They could discuss different factors such as the biology of the vectors in central Germany or land use.

The conclusions are not strong and are not connected with the objective of the work. I suggest re-write the conclusions.

Additional comments

Minor comments.

ABSTRACT:
Results: Blood samples from a total of 281 adult and nestlings...please indicate how many adults and how many nestlings.
Conclusion: The sentence "Currently there are only a few dates... of avian Haemosporidia" is long. I suggest reducing.

INTRODUCTION:
Lines 38-44: I suggest that the authors indicating names of parasites and the birds in the different examples.

MATERIAL AND METHODS:
Lines 78-82: Are correct the primers? Haem F2 and HaemR2L are the primers used for Leucocytozoon.
Lines 86-88: I suggest indicating why Blue tits nestling were only checked for Leucocytozoon and Trypanosoma and why great tits were not checked for T. avium.

RESULTS
Line 163: Is very interesting not detect Leucocytozoon in blood smears with a prevalence of 94.8% by PCR. Could the authors discuss deeper why this result in the discussion section?
Line 165: Why were excluded great tits? I suggest explaining it.
Line 173: How do you calculate the uncorrected p-distance between lineages?
Line 177: What lineages?
Lines 180-184: I suggest remove these sequences from the manuscript and indicate that 86 sequences could not be assigned due a bad quality and were removed from the analyses.

DISCUSSION
Line 217: What extra information gives the blood smears in this study? I suggest removing it because the information of the blood smears is null.
Line: 275: Culicidae.

CONCLUSION
Line 338-339: Because Plasmodium is avian malaria and Haemoproetus avian malaria-like, I suggest that in the line 339 Plasmodium must be indicated before Haemoproteus.

Reviewer 3 ·

Basic reporting

This paper reports on a survey of four genera of hematozoa from central Germany in six species of passerine birds. The rates of infection were generally high (except T. avium). Both microscopy and PCR were used in the survey.

The quality of the paper was generally adequate, though technical aspects of the writing could be improved throughout. Some of the reporting of results was not well directed toward questions being addressed, and questions that could be addressed using this dataset. For example, Figure 1 and Figure 2 were not informative, and the haplotype networks presented in subsequent figures contained much more effective visualizations of the same information. I strongly recommend deleting Figs. 1 and 2.

The most interesting part of the paper is the early Discussion that compares to previous results concerning the same host species elsewhere in Germany or Europe. This Discussion addresses the question of whether central Germany surveys such as this one are consistent with previous knowledge based on surveys of these host species from other locations. This section should heavily focus on the blue tit and great tit, which are the only thoroughly sampled host species in this paper. It should provide sample sizes for the previous studies to put the differences in prevalence in context for the reader. I suggest considering a table to specifically report these comparative results for tits, as this gets to the specific importance of this study.

The results for microscopy are not sufficiently reported or described. These data aren't documented anywhere that I could find. Are there microscopy photos archived somewhere? Why are these results not reported in the Summary_Individuals.xlsx document? Why are there no metadata available for that document? I could not find any information about the MalAvi or Genbank contributions of this paper. Nor could I find where the samples/extracts/smears may be archived.

The section about nestlings should be shortened way down, as it was tangential and failed to add to previous knowledge of the topic. The negative result should have been expected.

The paragraph beginning on Line 299 is vague and speculative and should be cut.

The last paragraph of Discussion, beginning on line 326 is not expressing any new ideas, and is built on a 'straw-man' argument that host-specificity of specific lineages is firmly predicted by the general tendencies of their genus. This paragraph should be deleted. Only the first sentence of it is valid, but that is also redundant with the Conclusions that follow. In the Conclusions, the phrase "currently only a few data..." is not true; though a more nuanced argument could be made that the diversity and geographic variation in haemosporidian communities warrant more study, even in Europe.

Line 275: Culicidae spelled incorrectly.
Line 290: "genera" should be "genus"

Experimental design

Adequate.

Validity of the findings

Valid.

---

## Round 0.2 · Minor Revisions

Please, follow all the corrections from Reviewer #1.

Reviewer 1 ·

Basic reporting

No comment

Experimental design

No comment

Validity of the findings

No comment.

Additional comments

Please find attached my few comments in the pdf file.

Annotated reviews are not available for download in order to protect the identity of reviewers who chose to remain anonymous.

Reviewer 2 ·

Basic reporting

no comment

Experimental design

no comment

Validity of the findings

no comment

Additional comments

Thank you for your revisions and reformatting of your paper entitled "Blood parasites in Passeriformes in central Germany: Prevalence and lineage diversity of Haemosporida (Haemoproteus, Plasmodium and Leucocytozoon) in six common songbirds". You have adequately addressed my comments regarding your paper. I particularly appreciate the changes you made in Material and Methods.

---

## Round 0.3 · accepted · Accept

All the comments raised the Reviewer were well addressed by the authors.

#